# Improving Composed Image Retrieval via Contrastive Learning with Scaling Positives and Negatives

## ABSTRACT

The Composed Image Retrieval (CIR) task aims to retrieve target images using a composed query consisting of a reference image and a modified text. Advanced methods often utilize contrastive learning as the optimization objective, which benefits from adequate positive and negative examples. However, the triplet for CIR incurs high manual annotation costs, resulting in limited positive examples. Furthermore, existing methods commonly use in-batch negative sampling, which reduces the negative number available for the model. To address the problem of lack of positives, we propose a data generation method by leveraging a multi-modal large language model to construct triplets for CIR. To introduce more negatives during fine-tuning, we design a two-stage fine-tuning framework for CIR, whose second stage introduces plenty of static representations of negatives to optimize the representation space rapidly. The above two improvements can be effectively stacked and designed to be plug-and-play, easily applied to existing CIR models without changing their original architectures. Extensive experiments and ablation analysis demonstrate that our method effectively scales positives and negatives and achieves state-of-the-art results on both FashionIQ and CIRR datasets. In addition, our methods also perform well in zero-shot composed image retrieval, providing a new CIR solution for the low-resources scenario. The code is released at https://anonymous.4open.science/r/45F4 and will be publicly available upon acceptance.

## CCS CONCEPTS

• **Information systems → Multimedia and multi-modal Retrieval; Image Search; Retrieval effectiveness**.

## KEYWORDS

composed image retrieval, contrastive learning

## 1 INTRODUCTION

Composed Image Retrieval (CIR) aims to retrieve images given a query composed of a modified text and a reference image. Unlike the standard text-to-image retrieval tasks, the modified text in CIR describes the unsatisfied attributes of the reference image or the new attributes based on the reference image. CIR provides a new idea for iteratively optimizing the retrieval results based on the

Unpublished working draft. Not for distribution.

Permission to make digital or hard copies of all or part of this work for personal or classroom use is granted without fee provided that copies are not made or distributed for profit or commercial advantage and that copies bear this notice and the full citation on the first page. Copyrights for components of this work owned by others than the author(s) must be honored. Abstracting with credit is permitted. To copy otherwise, or republish, to post on servers or to redistribute to lists, requires prior specific permission and/or a fee. Request permissions from permissions@acm.org.

*ACM MM, 2024, Melbourne, Australia*

© 2024 Copyright held by the owner/author(s). Publication rights licensed to ACM.
ACM ISBN 978-x-xxxx-xxxx-x/YY/MM
https://doi.org/10.1145/nnnnnnn.nnnnnnn

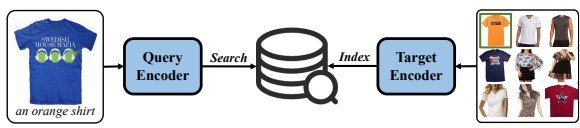

(a) Illustration of the Composed Image Retrieval (CIR) task

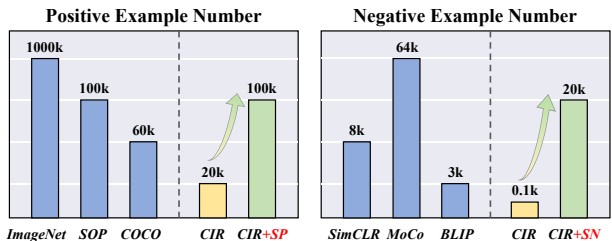

(b) Our proposed method effectively scales the number of positive and negative examples in the CIR task to a level comparable to other computer vision tasks and models.

**Figure 1: Task introduction and the motivation of this work.**

current text-to-image retrieval and thus has become a popular research task in the multi-modal field. Previous research on CIR typically involves model architecture [5, 9, 33] and optimization objectives [2, 5, 23, 35]. The methods for the model architecture focus on better representation and fusion methods for texts and images. The contribution of the works in this aspect includes (1) introducing vision-language pre-trained models, like CLIP [28], BLIP [18], as the backbone [2, 5, 23, 35] and (2) designing the novel late fusion [5, 33, 35] or early fusion [2, 16, 23] modules to fuse the reference image and the modified text to obtain the single query representation. Therefore, the popular model architecture in CIR can be illustrated in Fig.1(a), which consists of a query encoder and a target image encoder. In practice, a collection of image candidates is first converted into image representations by the image encoder for rapid indexing. When the user gives a reference image and a modified text, they are forwarded to the query encoder to fusion and obtain the query representation. Finally, the query representation is computed with the dot product or cosine similarity with all representations of candidate images, and the image with the top similarity is considered the target image.

The works for optimization objective [2, 35] focus on aligning the query representation with the target image representations. Advanced methods use contrastive learning [31] to align the representations of queries and images. The key to contrastive learning is to select correct and sufficient positives and negatives. The annotated triplets in the dataset, in the form of (*reference image, modified text, target image*), usually are regarded as positive examples, while negative examples are generated by replacing the target images with other ones in the mini-batch. However, as shown in

Fig.1(b), there are two challenges with these works: (1) The number of manually annotated triplets (20K) is deficient, leading to a lack of sufficient positive examples for the model. As a comparison, in other tasks using contrastive learning like visual representation learning [10], image-text retrieval [19, 27], and image retrieval [26], the number of positive examples is at least 60k; (2) Previous CIR tasks typically use in-batch negative sampling, with around 128 negative examples, while many successful works in contrastive learning use over 4k negative examples [7, 13, 18]. Existing works ignore these two problems at the data level, resulting in the inability of contrastive learning to fulfill its capabilities.

Therefore, this work is based on a universal and simple motivation: to scale the number of positive and negative samples of the CIR task to the same scale as other tasks with contrastive learning. To construct more positives for CIR, we propose a novel data generation method based on the multi-modal Large Language Model (MLLM). Specifically, we design a four-step pipeline to automatically construct positive samples, which includes (1) caption generation with MLLM; (2) reference-target image pair matching; (3) modified text generation based on templates; and (4) positive example construction. With the help of our method, plenty of acceptable positive examples can be generated without any manual annotation, scaling the triplet number from 20k to 100k without the use of external datasets (Fig.1(b)). To introduce more negatives for CIR, we design a two-stage fine-tuning framework. Specifically, in the first stage, we follow previous works [2, 4, 23, 35] and use in-batch negative sampling to enable the model to learn initial representation space for CIR; while in the second stage, we initialize the model trained in the first stage and freeze the target image encoder, only fine-tuning the query encoder. The frozen target image encoder introduces a large number of static representations of negatives at once (Fig.1(b)), guiding query encoders to optimize representation space rapidly. Note that the second stage has only about 1/20 of the time overhead of the first stage and can be easily superimposed on existing advanced models in CIR.

To verify the effectiveness of our method, we experiment extensively with both the full-supervised and zero-shot settings. For the full-supervised setting, we adopt our method in four advanced models in CIR with different backbones, achieving a 1%-6% performance improvement on the popular FashionIQ and CIRR datasets, reaching a new state-of-the-art. For the zero-shot setting, the model needs to be built without requiring human-labeled triplets for training. We apply our method to in-domain and out-of-domain image datasets to construct sufficient positives and negatives for CIR. With fewer image scales than the baselines, the superior performance of our method demonstrates the ease with which our method can be applied to low-resource scenarios.

The contributions of our paper can be summarized as follows:

- We propose a data generation method with the multi-modal large language model to scale positive examples in CIR, which can automatically build high-quality positive examples based on image datasets only.
- We propose a two-stage plug-to-play framework to scale negative examples during fine-tuning, whose second stage can be quickly adapted to almost any model in CIR with 1/20 time overhead of the first stage.

- Extensive experiments and analysis under the full-supervised and zero-shot setting demonstrate the effectiveness and superiority of our proposed method, which achieves state-of-the-art performance on both FashionIQ and CIRR datasets.

## 2 RELATED WORK

*Composed Image Retrieval.* The recent paradigm in CIR [4, 33, 40, 41] consists of three main steps: (1) extracting the representation of both images and sentences; (2) fusing the representations of sentences and reference images to obtain query representations; (3) aligning the representations of queries and target images with similar semantics. For the first step, early models in CIR utilize two separate encoders [11, 14, 33, 38] while recent CIR models [4, 21, 35, 41] exploit pre-trained vision-language encoders [25, 28] as the backbone. Some works [4, 41] simply use the global representations extracted from these pre-trained encoders while other works [35, 38, 40] integrate local and global representations. For the second step, some works [4, 33, 34, 41] leverage weights or gating mechanisms, while other works [38, 40] design combining modules like cross-modal transformer. For the third step, the most commonly used loss functions in CIR are triplet loss [8, 30, 40], contrastive learning [4, 15, 31, 33–35, 41]. Recent advanced methods in the CIR predominantly employ a combination of dual encoders and contrastive learning with in-batch negative sampling. We treat the models obtained from these methods as the first-stage models and continue to train them in the second stage to improve CIR performance further.

*Data Generation for CIR.* InstructPix2Pix [6] first uses GPT-3 to generate modified text for captions and then utilizes a diffusion model to generate images for these texts. COVR [32] mine similar captioned videos from a large database and use a language model to generate modified text that describes the differences between the videos, resulting in the WebVid-CoVR dataset with 1.6 million triplets. CASE [16] uses a data roaming approach that rephrases labels from a large-scale VQA dataset into a form suitable for composed image retrieval. CompDiff [12] constructs triplets for CIR datasets by automatically generating modified texts and corresponding images using large language models and diffusion models. Unlike these works that often require generating images or well-labeled datasets, our method is built on a real image collection without the need for any additional manual annotation and leverages the capabilities of MLLM to construct triplets.

*Negative Sampling in Contrastive Learning.* In the realm of contrastive learning, negative sampling techniques have evolved to enhance model performance: the in-batch negative sampling from SimCLR [7] selects negative examples from the same batch, while the memory bank approach in Bank [37] utilizes a stored set of past instances for more diverse negatives. Additionally, MoCo [13] employs a moving average of representations to create dynamic negatives, contributing to robust representation learning. Compared to Memory Bank and MoCo, our method does not dynamically update the negatives with the aid of additional queues or momentum encoders; instead, it fine-tunes the model for the second stage by introducing a large number of static negative samples at once.

# 3 METHOD

## 3.1 Preliminary

Suppose a CIR dataset consists of $N$ annotated triplets, where the $i^{\text{th}}$ triplet $x_i$ is denoted as

$$x_i = (r_i, m_i, t_i), r_i, t_i \in \Omega, m_i \in T \quad (1)$$

where $r_i$, $m_i$, and $t_i$ represent the reference image, the modified text[1], and the target image of the $i^{\text{th}}$ example, respectively, while $\Omega$ is the candidate image set containing all reference and target images of the triplets and $T$ is the text set containing all modified texts. The CIR task aims to use the reference image $r_i$ and the modified text $m_i$ to compose a query $q_i$, and retrieve the target image $t_i$ from the candidate set $\Omega$ with $q_i$.

Then, we describe the classical paradigm of CIR. Multiple annotated triplets are combined into a mini-batch, and the reference images and modified texts in the same batch are then encoded using a query encoder $F(\cdot)$ query representations. The target images are encoded using an image encoder $G(\cdot)$ to obtain target image representations. For simplicity, we rewrite the representations for the triplet $(r_i, m_i, t_i)$ as $\mathbf{q}_i = F(r_i, m_i)$ and $\mathbf{t}_i = G(t_i)$, respectively. The cosine similarity $f(\cdot, \cdot)$ is then adopted to calculate the similarity between the query and target image representations. Recall that current methods based on contrastive learning usually treat the annotated examples as positive examples and treat the examples obtained by replacing the target image in the positive examples with another image in the mini-batch as the negatives. Then contrastive learning is used to pull the query representations and target image representations in positive examples closer while pushing query representations and target image representations in negative examples further, which can be expressed as

$$\mathcal{L}_{\text{cl}}^{\text{t}} = \frac{1}{B} \sum_{i=1}^{B} -\log\left(\frac{\exp(f(\mathbf{q}_i, \mathbf{t}_i)/\tau)}{\sum_{j=1}^{B} \exp(f(\mathbf{q}_i, \mathbf{t}_j)/\tau)}\right) \quad (2)$$

where $B$ is the batch size and $\tau$ is a temperature hyper-parameter.

Despite the good results achieved with this current paradigm, the lack of negative and positive examples still severely limits the performance of contrastive learning. To address these problems, we first propose a method of scaling positive examples using a multi-modal large language model (MLLM). Then, we investigate the impact of different types of negative examples on CIR performance and find that using negative examples obtained by replacing the target image is simple and most effective. Therefore, we propose a two-stage fine-tuning strategy, scaling negative examples using a caching technique based on existing models.

## 3.2 Scaling Positive Examples

Due to the high cost of manually labeling triplets, we propose a simple but effective method with a multi-modal Large Language Model (MLLM) to construct the triplets for CIR. As shown in Fig.2, given an image dataset[2] $D = \{I_1, I_2, ..., I_M\}$ with size $M$, our method consists of four steps: (1) Generating a suitable caption for each image to obtain the image-text pairs; (2) Constructing M (reference

image, target image) pairs; (3) Generating modified texts for image pairs using the captions; (4) Combining the modified texts and image pairs to form triplets.

*Caption Generation.* We introduce a MLLM $g_{\text{mllm}}(\cdot, \cdot)$ to generate a corresponding caption for each image in the dataset. Specifically, we design a prompt template $P_{\text{cap}}(type, k)$ to guide the MLLM to obtain a brief caption for each image under constrained conditions, where $type$ and $k$ are two dataset-specific parameters to simulate the type and length of modified text in the real dataset. For an image $I_i$ in the candidate image set, we input $I_i$ and $P_{\text{cap}}$ together into the MLLM to obtain the corresponding caption $C_i$:

$$C_i = g_{\text{mllm}}(I_i, P_{\text{cap}}(type, k)). \quad (3)$$

Then we can obtain M image-text pairs $\{(I_1, C_1), ..., (I_M, C_M)\}$. In practice, the $P_{\text{cap}}$ used in this work is written as follows:

```
Please briefly describe the {type} in {k} words.
```

*Image Pair Match.* After obtaining the image-text pair, we need to match two image-text pairs to generate a quadruplet. Regarding the image in an image-text pair as the reference image, the naive method randomly chooses the image from another image-text pair as the target image. However, a randomly selected target image may be too similar to the reference image to construct precise modified text or too dissimilar to help models improve performance. Therefore, we introduce a uni-modal image encoder $g_{\text{img}}(\cdot)$ to get the representation of every image and calculate the pairwise similarity between two different images $I_i$ and $I_j$:

$$sim_{ij} = f(g_{\text{img}}(I_i), g_{\text{img}}(I_j)) \ (1 \leq i, j \leq M, i \neq j) \quad (4)$$

Then we can rank the similarities related to $I_i$ in descending order. Only one image whose similarity rank is between $[c_0, c_1)(c_0 < c_1)$ will be chosen as the target image, where $c_0$ and $c_1$ are two hyper-parameters. In practice, we regard each image in the dataset $D$ as the reference image and sample a target image for each image. We denote the target image for image $I_i$ as $I_i^{\text{t}}$, therefore, we can get $M$ (reference image, target image) pairs $\{(I_1, I_1^{\text{t}}), ..., (I_M, I_M^{\text{t}})\}$. We combine these image pairs with their corresponding captions to form M quadruplets:

$$\{(I_1, C_1, I_1^{\text{t}}, C_1^{\text{t}}), ..., (I_M, C_M, I_M^{\text{t}}, C_M^{\text{t}})\} \quad (5)$$

*Modified Text Generation.* Given one quadruplet $(I_i, C_i, I_i^{\text{t}}, C_i^{\text{t}})$ by the last step, we use a prompt template $P_{\text{temp}_k}(k \in \{0, 1, 2\}$ to form a modified text $m_i^{\text{temp}_k}$:

$$m_i^{\text{temp}_k} = P_{\text{temp}_k}(C_i, C_i^{\text{t}}) \quad (6)$$

In this work, we consider three types of templates below.

```
P_temp₀: {Cᵢᵗ} instead of {Cᵢ}
P_temp₁: Unlike {Cᵢ}, I want {Cᵢᵗ}
P_temp₂: {Cᵢᵗ}
```

Note that we attempt to use LLM to post-process the generated modified text. The first method involves using LLM to make the

---

[1] In this work, we refer to the text in the CIR triplet as a "modified text", which is also referred to as a "modification sentence" or "modification text" in other works.

[2] Image dataset here could be $\Omega$ in the CIR dataset or any image dataset.

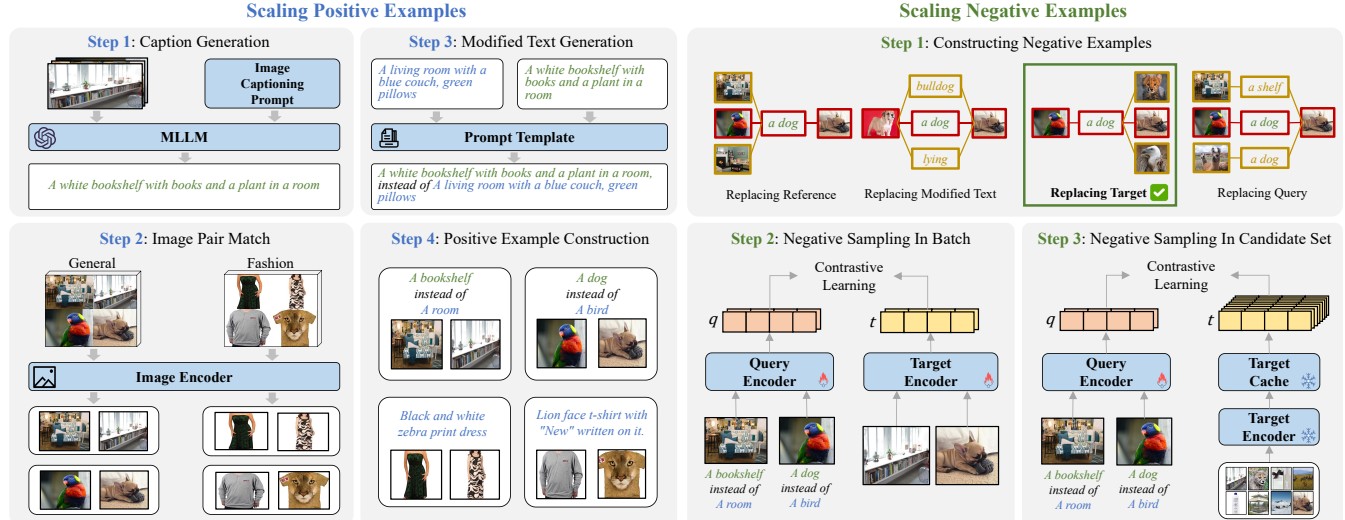

**Figure 2: Overview of Our Framework of Scaling Positive Examples and Negative Examples. We abbreviate some of the modified texts due to space constraints.**

modified text more diverse and fluent. The second method uses in-context learning to make LLM mimic the modified text in annotated datasets. However, based on our experiments, neither of these methods surpasses the prompt template method. Specific results can be found in supplementary materials.

*Positive Example Construction.* Finally, we could combine image pairs from the second step with the modified texts obtained in the third step to get new M triplets $\{(I_i, m_i^{temp}, I_i^t)\}$. So, we can obtain an expanded dataset that is comparable in size to the original dataset. We could use these new examples as a complement to the annotated dataset. We could also use these examples to train a model from scratch, thus allowing for fully automated training of a CIR model without human involvement.

### 3.3 Scaling Negative Examples

Recent works in visual contrastive representation learning [13, 37] have shown that scaling negative numbers can effectively improve performance. However, existing works in CIR employ in-batch negative sampling strategies, restricting the model from seeing enough negatives. Furthermore, recalling that the labeled data in CIR is a triplet, it is theoretically possible to construct negative examples by replacing any element in the triplet. Most works [2, 5, 33, 35] only use the "replace the target image" strategy to construct negative samples without additional interpretation. Therefore, we first explore the performance impact of different methods of constructing negative examples and find that "Replacing the target image" leads to more true and hard negatives than other methods with the popular CIR datasets. After determining the method of negative example construction, we propose a two-stage fine-tuning strategy for CIR that leverages a two-stage framework to scale negative examples during fine-tuning.

*Constructing Negative Examples.* Considering the annotated data in CIR are triplets, for triplet $(r_i, m_i, t_i)$, there are four methods

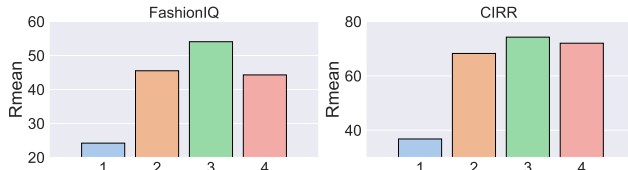

**Figure 3: Performance of four different methods on negative example construction. The number on the horizontal axis corresponds to the serial number before the different replacing methods in Section 3.3.**

of negative example construction by randomly sampling another triplet $(r_j, m_j, t_j)$:

(1) Replacing the reference image, obtaining $(r_j, m_i, t_i)$;
(2) Replacing the modified text, obtaining $(r_i, m_j, t_i)$;
(3) Replacing the target image, obtaining $(r_i, m_i, t_j)$;
(4) Replacing the whole query pair, i.e. the reference image and modified text, obtaining $(r_j, m_j, t_i)$.

Most previous works use only the third method [2, 5, 9, 23, 33, 35], and Wang et al. [34] uses the first three methods jointly. However, none of the existing works have explored all four methods completely. To this end, we compare these four negative construction methods while other settings remain the same. As shown in Fig.3, we find that constructing the negative examples by replacing target images works best. Based on the examples in Fig.2, we can observe that the other three methods easily generate relatively simple or false negatives. For example, since some modified texts (e.g., "a dog") only describe the target image, replacing the reference image with another image can lead to false negatives. Similarly, if the reference image is very similar to the target images, this type of data leads the model to directly use the reference image to retrieve the

target image, making it easy to generate false negatives when replacing the modified text (e.g., "dog" and "lying"). Lastly, replacing the whole query pair leads to the simple negatives as the reference image and modified text significantly differ from those in the positive example (e.g., "sofa+shelf" and "llama+dog"). Compared with the other three methods, "replacing the target image" is inherently aligned with the final application scenario, and the probability of generating false negatives is relatively low. In the supplementary materials, we report the performance of every combination of four types of negative examples. The experimental results suggest that incorporating other types of negative examples may lead to increased overhead and potentially compromise model performance. For this reason, we keep consistent with previous work and only consider the negative example type of replacing the target image.

*Two-Stage Fine-tuning.* Previous work on extended negative examples, such as Memory Bank [37] and MoCo [13], has focussed on visual representation learning, whose models typically follow a simple Siamese network architecture. However, CIR tasks require an information fusion of visual and language, and different methods follow different backbones, such as CLIP [28], BLIP [18], and BLIP-2 [17]. Therefore, we propose a more general two-stage framework to ensure fast adaptation of different models in CIR (Right half of Fig.2). Specifically, in the first stage, we fine-tune both the query encoder and the target encoder with in-batch negative sampling as in Eqn.2 following previous works [2, 5, 23]; while in the second stage, we freeze the target encoder and only fine-tune the query encoder. Therefore, all candidate images, i.e., the entire $\Omega$, can advance past the frozen target image encoder, cached before the second fine-tuning stage. Finally, for triplet $(m_i, r_i, t_i)$, we utilize all non-target images from the candidate set, i.e. $\Omega - \{t_i\}$, to form negative examples. The contrastive loss for the second stage can be expressed as

$$\mathcal{L}_{\text{cl}}^{\text{2rd}} = \frac{1}{B} \sum_{i=1}^{B} -\log \frac{\exp(f(\mathbf{q}_i, \hat{g}(t_i))/\tau)}{\sum_{t_j \in \Omega} \exp(f(\mathbf{q_i}, \hat{g}(t_j))/\tau)} \quad (7)$$

where $\hat{g}(.)$ represent the frozen target image encoder. It is worth noting that the second stage is very efficient. According to our estimates on different baselines, one training epoch on average takes 12 minutes, and typically 50 epochs are needed for the first stage, resulting in a total duration of around 10 hours. While in our additional second stage, pre-computing representations take an average of 10 minutes, with each epoch taking 5 minutes, and only 5 epochs are required, around half an hour in total.

## 4 EXPERIMENTS

### 4.1 Experimental Setup

*4.1.1 Baselines.* To evaluate the superiority of our method, we conduct experiments on four advanced models in CIR: TG-CIR [35], CLIP4CIR [5], BLIP4CIR [23] and SPRC [2].

**TG-CIR** [35] uses $\text{CLIP}_{\text{ViT-B/16}}$ as the backbone, which exploits the global and local attribute representations and information from the target image to guide both query fusion and metric learning.

**CLIP4CIR** [5] uses $\text{CLIP}_{\text{ResNet50x4}}$ as the backbone, which simply regards element-wise sum as a fusion approach.

**BLIP4CIR** [23] uses $\text{BLIP}_{\text{base}}$ as the backbone, which adopts the fusion encoder of BLIP to fuse the reference image tokens and modified text tokens. We do not include an extra re-ranker to ensure the evaluation protocols are consistent.

**SPRC** [2] uses $\text{BLIP-2}_{\text{pretrained-vitl}}$ as the backbone, which exploits the QFormer [17] as an encoder for query and target image sharing.

*4.1.2 Training Protocol.* We directly use checkpoints released by baseline works as the first stage models to avoid retraining. For the second stage, we calculate image representations for all images before training and only finetune the query encoder using scaled positives and negatives for 5 epochs. For the main results in section 4.2, we only use images in $\Omega$, so it can be fairly compared to any model that uses the original dataset.

**Table 1: Average token length calculated by LLAVA Tokenizer [20] of the modified text and triplet count statistics for the annotated and generated training sets.**

| Dataset | Annotated | | Generated | |
|---------|-----------|----------|-----------|----------|
| | Token | Triplet | Token | Triplet |
| FashionIQ [36] | 7.8 | 18k | 16.5 | 96k |
| CIRR [21] | 15.4 | 28k | 20.9 | 128k |

*4.1.3 Evaluation Datasets.* We evaluate our model on two commonly used CIR datasets: FashionIQ [36] and CIRR [21].

**FashionIQ** [36] consists of 30,134 examples extracted from 77,684 images crawled from fashion websites. These images are categorized into Dress, Shirt, and Top&Tee. The modified text is manually annotated for each pair of reference and target images. As in [4, 40, 41], we use 18,000 examples for training and 6,016 validation examples for testing since the "real" test set is unavailable.

**CIRR** [21] (Composed Image Retrieval on Real-life images) contains 21,552 real-life images from the web taken from $NLVR^2$, a popular natural language reasoning dataset. CIRR contains 36,554 examples, of which 28,225 examples are used for training, 4,181 for validation, and 4,148 for testing. In addition, the images in this dataset are divided into several semantically similar groups to evaluate $R_{\text{subset}}@K$ metric (see below).

*4.1.4 Evaluation Metrics.* Recall@K ($R@K$) is the proportion of queries for which the retrieved top K images include the correct target image. $Recall_{\text{subset}}@K$ ($R_{\text{subset}}@K$) is nearly the same as $R@K$ but the model only retrieves inside the semantically similar group of the reference image. For the FashionIQ dataset, following previous works [2, 4, 23], we evaluate our model through $R@K$ ($K = 10, 50$) on the original protocol. As in [41], we also report the mean of all $R@K$ scores as Rmean. For the CIRR dataset, following previous works [4, 22], we evaluate our model through $R@K$ ($K = 1, 5, 10, 50$) and $R_{\text{subset}}@K$ ($K = 1, 2, 3$). As in [2], we also report ($R@5+R_{\text{subset}}@1$)/2 as Rmean.

*4.1.5 Implementation Details.* We use LLaVA-v1 [20] as a multimodal large language model for caption generation. We use the advanced model $\text{unicom}_{\text{ViT-L/14}}$ [1] for the unimodal image encoder. As used by CLIP4CIR [5], we leverage AdamW [24] optimizer. All experiments are conducted on a single Tesla V100 GPU.

**Table 2: Evaluation results of various models on FashionIQ. The best results are in boldface.**

| Methods | Backbone | Dress | | Shirt | | Top&Tee | | Average | | |
|---|---|---|---|---|---|---|---|---|---|---|
| | | R@10 | R@50 | R@10 | R@50 | R@10 | R@50 | R@10 | R@50 | Rmean |
| CIRPLANT [21] | w/o VLP | 17.45 | 40.41 | 17.53 | 38.81 | 21.64 | 45.38 | 18.87 | 41.53 | 30.20 |
| ARTEMIS [9] | w/o VLP | 27.16 | 52.40 | 21.78 | 43.64 | 29.20 | 54.83 | 26.05 | 50.29 | 38.17 |
| ComqueryFormer [39] | w/o VLP | 28.85 | 55.38 | 25.64 | 50.22 | 33.61 | 60.48 | 29.37 | 55.36 | 42.37 |
| PL4CIR [41] | CLIP | 33.60 | 58.90 | 39.45 | 61.78 | 43.96 | 68.33 | 39.02 | 63.00 | 51.01 |
| TG-CIR [35] | CLIP | 35.55 | 59.44 | 40.24 | 62.37 | 43.65 | 67.36 | 39.81 | 63.06 | 51.44 |
| +SPN | CLIP | 36.84 | 60.83 | 41.85 | 63.89 | 45.59 | 68.79 | 41.43 | 64.50 | 52.97 |
| CLIP4CIR [4] | CLIP | 38.18 | 62.67 | 44.01 | 64.57 | 45.39 | 69.56 | 42.52 | 65.60 | 54.06 |
| +SPN | CLIP | **38.82** | **62.92** | **45.83** | **66.44** | **48.80** | **71.29** | **44.48** | **66.88** | **55.68** |
| BLIP4CIR [23] | BLIP | 44.22 | 67.08 | 45.00 | 66.68 | 49.72 | 73.02 | 46.31 | 68.93 | 57.62 |
| +SPN | BLIP | **44.52** | **67.13** | **45.68** | **67.96** | **50.74** | **73.79** | **46.98** | **69.63** | **58.30** |
| SPRC [2] | BLIP-2 | 49.18 | 72.43 | 55.64 | 73.89 | 59.35 | 78.58 | 54.92 | 74.97 | 64.85 |
| +SPN | BLIP-2 | **50.57** | **74.12** | **57.70** | **75.27** | **60.84** | **79.96** | **56.37** | **76.45** | **66.41** |

**Table 3: Performance comparison of various models on CIRR. The best results are in boldface.**

| Methods | Backbone | Recall@K | | | | $R_{subset}$@K | | | Rmean |
|---|---|---|---|---|---|---|---|---|---|
| | | K=1 | K=5 | K=10 | K=50 | K=1 | K=2 | K=3 | |
| CIRPLANT [21] | w/o VLP | 19.55 | 52.55 | 68.39 | 92.38 | 39.20 | 63.03 | 79.49 | 45.88 |
| ARTEMIS [9] | w/o VLP | 16.96 | 46.10 | 61.31 | 87.73 | 39.99 | 62.20 | 75.67 | 43.05 |
| ComqueryFormer [39] | w/o VLP | 25.76 | 61.76 | 75.90 | 95.13 | 51.86 | 76.26 | 89.25 | 56.81 |
| TG-CIR [35] | CLIP | 45.23 | 78.34 | 87.13 | 97.30 | 72.84 | 89.25 | 95.13 | 75.59 |
| +SPN | CLIP | **47.28** | **79.13** | **87.98** | 97.54 | **75.40** | **89.78** | 95.21 | **77.27** |
| CLIP4CIR [4] | CLIP | 42.80 | 75.88 | 86.26 | 97.64 | 70.00 | 87.45 | 94.99 | 72.94 |
| +SPN | CLIP | **45.33** | **78.07** | **87.61** | **98.17** | **73.93** | **89.28** | **95.61** | **76.00** |
| BLIP4CIR [23] | BLIP | 44.77 | 76.55 | 86.41 | **97.18** | 74.99 | 89.90 | 95.59 | 75.77 |
| +SPN | BLIP | **46.43** | **77.64** | **87.01** | 97.06 | **75.74** | **90.07** | **95.83** | **76.69** |
| SPRC [2] | BLIP-2 | 51.96 | 82.12 | 89.74 | 97.69 | 80.65 | 92.31 | 96.60 | 81.39 |
| +SPN | BLIP-2 | **55.06** | **83.83** | **90.87** | **98.29** | **81.54** | **92.65** | **97.04** | **82.69** |

We manually tune $\tau \in \{0.01, 0.02, 0.03, 0.05\}$ and $learning\_rate \in \{2e-6, 5e-6, 6e-6, 1e-5, 2e-5\}$. Detailed hyper-parameters are reported in the supplementary materials.

We analyze the modified text in the two datasets using LLAVA tokenizer [20] and count the average annotated token length in Table 1. For FashionIQ, we set *type* to the name of the split, i.e., dress/shirt/top tee, and $k$ to 5. For CIRR and Conceptual Caption, we set *type* to "image" and $k$ to 10 in the image captioning template. The detailed data statistics for both the generated and annotated triplets are provided in Table 1. For both datasets, we set $c_0$ to 10000. We set $c_1$ to 20000 for FashionIQ and 15000 For CIRR.

## 4.2 Main Results

We compare our method against the following baseline methods: CIRPLANT [21], ARTEMIS [9], ComqueryFormer [39], PL4CIR [41], TG-CIR [35], CLIP4CIR [4], BLIP4CIR [23], SPRC [2]. Details about these models can be found in supplementary materials. We abbreviate the method of scaling positive examples as **SP**, the method of

scaling negative examples as **SN**, and the superposition of the two methods as **SPN**.

*Results on FashionIQ.* Table 2 illustrates the comparison between our model and other recent studies on FashionIQ. It demonstrates that our plug-and-play approach improves the effectiveness of all four baseline models with different architectures. SPN boosts the R@10 metric for TG-CIR by 3.8%, CLIP4CIR by 4.1%, BLIP4CIR by 1.5%, and SPRC by 2.6%. SPN enhances the R@50 of TG-CIR by 3%, CLIP4CIR by 2%, BLIP4CIR by 1%, and SPRC by 2%. This mainly benefits from more negative and positive examples in contrastive learning, which allows the model to learn a better representation.

*Results on CIRR.* Table 3 illustrates the comparison between our model and other recent studies on CIRR. It shows that SPN also improves the performance of all four baseline models. SPN increases the R@1 of TG-CIR by 4.5%, CLIP4CIR by 5.9%, BLIP4CIR by 3.7%, and SPRC by 6%. SPN improves the R@5 of TG-CIR by 1%, CLIP4CIR by 2.9%, BLIP4CIR by 1.4%, and SPRC by 2.1%. This proves our

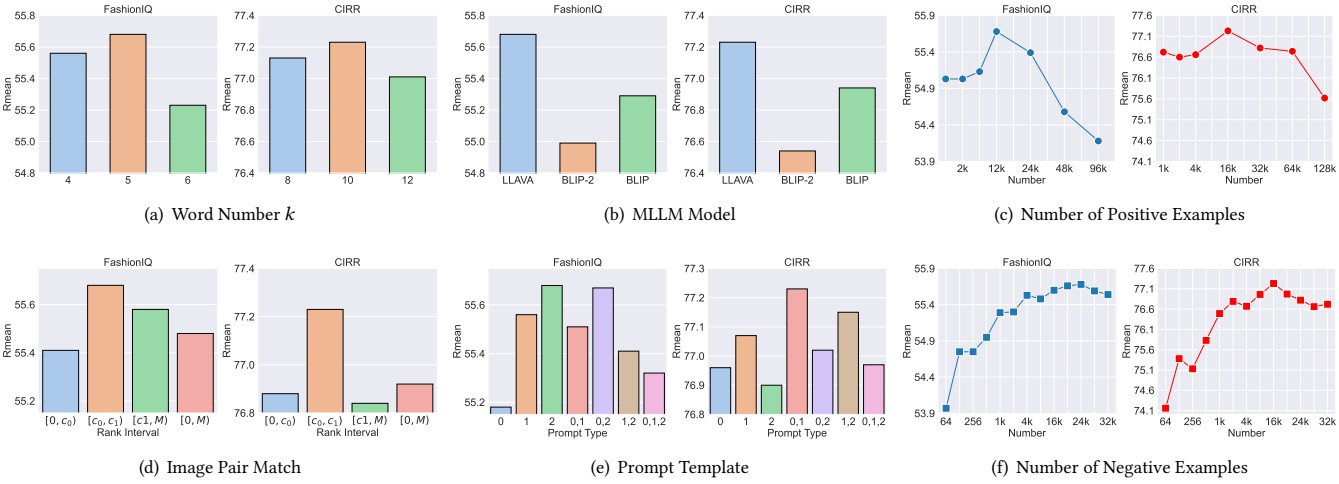

Figure 4: Discussion of the core components in the method. The results shown in the figures are on the validation set.

method works well for images in both general and fashion scenes. SPN promotes the $R_{subset}@1$ of TG-CIR by 3.5%, CLIP4CIR by 5.6%, BLIP4CIR by 1%, and SPRC by 1.1%. The objective of the subset test is to justify whether the model can distinguish between hard negative examples [21]. Such a boost indicates that our model can learn more fine-grained representations than the base model, thus distinguishing harder negative examples.

Table 4: Ablation results on CLIP4CIR.

| Model | FashionIQ | | CIRR | | |
|---|---|---|---|---|---|
| | R@10 | R@50 | R@1 | R@5 | $R_{subset}@1$ |
| CLIP4CIR | 42.52 | 65.60 | 43.96 | 77.68 | 70.84 |
| +**SP** | 43.83 | 66.66 | 44.75 | 79.45 | 72.85 |
| +**SN** | 43.43 | 66.45 | 46.35 | 79.67 | 73.07 |
| +**SPN** | **44.48** | **66.88** | **46.97** | **80.29** | **74.17** |

## 4.3 Ablation Study

*Contribution of SP and SN.* To evaluate the effectiveness of SP and SN, we train CLIP4CIR using several variants of our method and test on the validation set of CIRR and FashionIQ. SP variant conducts contrastive learning with in-batch negative sampling on the scaled positive examples. SN variant only scales negative examples without exploiting new positive examples. The results illustrate that removing either SN or SP significantly decreases performance. SP and SN can improve the baseline model by 1.3% to 3.1% on the two datasets, respectively. SN is more effective for the CIRR dataset. SP is more useful for the FashionIQ dataset. We attribute this phenomenon to the fact that the modified texts are more complex in the CIRR dataset and that contrast learning is more lacking in negative than positive examples. While the modified texts are simple in FashionIQ, the situation is exactly the opposite.

*Discussion on $k$.* Since the LLAVA tokenizer utilizes the BPE tokenization method, which typically results in a word count to token count ratio of 1:2. So the corresponding word counts for FashionIQ are around 4, and for CIRR, they are around 8. Therefore, we experiment with $k$ values that approximate the word count of the modified text in the annotated triplets. As shown in Fig.4(a), we find that slightly exceeding the annotated word count yields better results, as lower or higher values lead to performance degradation.

*Discussion on MLLM Model.* The MLLM we use can be replaced with any model that can generate captions for images, so we try three representative models LLAVA [20], BLIP [18], and BLIP-2 [17] in Table 4(b). We find that LLAVA, with great instruction fine-tuning, works best among the three models. But surprisingly, BLIP works better than BLIP-2. This suggests that BLIP-2's ability to follow image captioning instructions is not very good. At the same time, using different MLLMs consistently yields better results than w/o SP, indicating that our method is insensitive to different MLLMs.

*Discussion on Number of Positive Examples.* SP allows for constructing many triplets based on images, so we consider exactly how many additional triplets on top of the existing ones work best. As shown in Fig 4(c), as the number of positive examples rises, the effect of the model increases and then decreases, with the best results when increasing nearly 60% of the number of original triplets, that is 12k for FashionIQ and 16k for CIRR.

*Discussion on Image Pair Match.* In Fig.4(d), we explore four methods for constructing image pairs. The first method involves selecting target images with the highest similarity to the reference image. The second method entails choosing target images with moderate similarity to the reference image. The third method focuses on selecting target images with the lowest similarity to the reference image, while the fourth method involves selecting target images randomly from the entire set. Our findings indicate that the second method consistently produces superior results across both datasets.

**Table 5: Zero-shot results.**

| Model | FashionIQ | | CIRR | | |
|---|---|---|---|---|---|
| | R@10 | R@50 | R@1 | R@5 | $R_{subset}$@1 |
| *Out-of-Domain Image Dataset* | | | | | |
| CLIP [28] | 19.04 | 35.03 | 12.65 | 38.41 | 34.29 |
| PIC2WORD [29] | 24.70 | 43.70 | 23.90 | 51.90 | - |
| SEARLE-OTI [3] | 27.61 | 47.90 | 24.87 | 52.31 | 53.80 |
| **SPN-CC** | **28.97** | **49.54** | **34.34** | **65.42** | **64.87** |
| *In-Domain Image Dataset* | | | | | |
| **SPN-IN** | **31.11** | **52.19** | **36.55** | **67.69** | **67.28** |

This observation can be attributed to the higher quality of triplets generated through this selection approach compared to the others.

*Discussion on Prompt Templates.* We can combine three prompt templates in 7 ways. For two or more combinations of templates, we obtain a corresponding number of modified texts for each triplet and randomly select one during training. As shown in Fig.4(e), we find that for CIRR, a mixture of the first two works best. For FashionIQ, only the third works best. This indicates that in FashionIQ, more modified texts directly describe the target image.

*Discussion on Number of Negative Examples.* Because SN could exploit many images as negative examples, an experiment is conducted to verify the relationship between the number of negative examples and the performance. As shown in Fig 4(f), the model performs better as the number of negative examples rises and works best when all images in the candidate image set are used as negative examples, which is 24k for FashionIQ and 16k for CIRR. We additionally scale negative examples with images from external MSCOCO datasets. However, we observe a decline in performance.

### 4.4 Results of Zero-Shot CIR

Zero-shot CIR is aimed at building a CIR model without requiring human-labeled triplets for training [29]. For comparison under the zero-shot setting, we introduce two advanced baselines for zero-shot CIR, PIC2WORD [29] and SEARLE [3]. Following these two baselines, we use $CLIP_{ViT-L/14}$ as the backbone. Before training, we use the method described in Section 3.2 to generate a CIR dataset from an image dataset. Then contrastive learning with in-batch negative sampling is used for first-stage fine-tuning and the method described in Section 3.3 is used to scale negative examples for second-stage fine-tuning.

Neither of these baselines uses an in-domain image dataset for training. Therefor, we also utilize images from the out-of-domain dataset Conceptual Caption (CC3M), comprised of 3.3 million image-caption pairs from the Internet, to generate positive examples for a fair comparison with PIC2WORD. Specifically, we randomly select the 50k images in CC3M to construct the CIR dataset due to computing resource limitations. The number of 50k images is equal to 1.7% of that PIC2WORD used and 50% of that SEARLE used. We abbreviate our model trained with this setting as **SPN-CC**. As shown in Table 5, **SPN-CC** gets the best results while using the

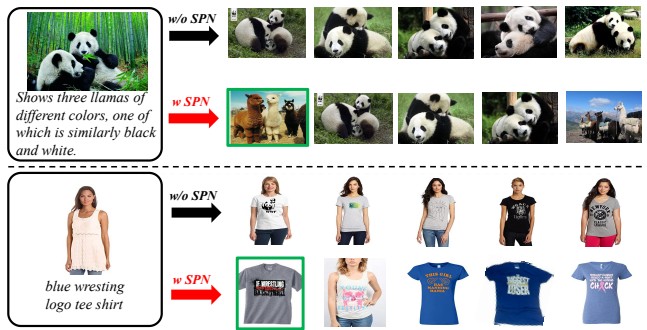

**Figure 5: Comparison of retrieval results between the CLIP4CIR model w/o and w SPN.**

least amount of images. This suggests that, given a collection of images out of the domain, our method can automatically construct appropriate triplets and train an acceptable model in the zero-shot setting. We also explore the setting of in-domain images, i.e., those in FashionIQ and CIRR, and abbreviate the model trained with this setting as **SPN-IN**. Under this setting, SPN-IN yields better results than SPN-CC using out-of-domain data. This suggests that if in the future we need to do a composed image retrieval task for a new scene but with few labeling costs, an accepted solution is to use our SPN method to automatically construct the positive examples within this scene and train a model from scratch.

### 4.5 Case Study

Fig.5 indicates the retrieval cases of CLIP4CIR w/o and w SPN. The first example is selected from CIRR, and the second one is from FashionIQ. For both examples, we can find that using SPN allows us to learn more of the **rarer concepts** (e.g. "llama", "logo"), thus enhancing the base model. In the meantime, we can find that the base model has difficulty in retrieving the correct image when the **reference and target images are very different** (e.g., "panda" and "llama", "dress" and "tee"), and SPN narrows this gap. More examples can be found in supplementary materials.

## 5 CONCLUSION

The Composed Image Retrieval (CIR) task uses a composed query to retrieve target images. While existing methods have achieved impressive results, limited labeled data and contrastive learning with in-batch negative sampling limit the performance of their methods. To address these problems, we first propose a data generation method using a multi-modal large language model to scale positives. We then propose a two-stage fine-tuning framework to scale negatives, introducing static representations of negatives in the second stage. These improvements are plug-and-play, enhancing existing CIR models without architecture changes. Extensive experiments show that we obtain state-of-the-art results on the FashionIQ and CIRR datasets. Moreover, our method could be applied to zero-shot composed image retrieval, offering a novel solution for unannotated CIR scenarios.

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
