# OpenReview forum: "Improving Composed Image Retrieval via Contrastive Learning with Scaling Positives and Negatives"
_acmmm.org/ACMMM/2024/Conference — MM2024 Poster_

### Official Review · Reviewer_hQWV · 2024-05-21

**Rating:** 3
**Confidence:** 3

**Summary:**

In this paper, the authors focused on the task of composed image retrieval (CIR), where the query consists of a reference image and a modified text. The authors found that the existing CIR datasets lack positive examples and negative examples. Toward this end, the authors introduced a MLLM-based positive examples generation method and a two-stage fine-tuning strategy. The proposed framework can be plug-and-play to the existing CIR methods.

**Strengths:**

There are three strong points:
1.	The authors proposed a data generation method with MLLM to augment the CIR datasets.
2.	The designed two-stage plug-and-play framework can scale up negative examples efficiently.
3.	Experiments prove the effectiveness of the proposed method.

**Limitations:**

There are also some weak points:
1.	The primary contribution appears to be centered solely on expanding the scale of the data, which may be considered somewhat lacking.
2.	The modified texts within the generated triplets represent descriptions of the target image, potentially deviating from the original data format. Furthermore, other related works, such as LIMN+[1] and BLIP4CIR[2], aim to address the data scarcity issue. However, the manuscript lacks an analysis of the advantages of the proposed method compared to other data generation approaches.
[1] Self-Training Boosted Multi-Factor Matching Network for Composed Image Retrieval. TPAMI 2024.
[2] Bi-directional Training for Composed Image Retrieval via Text Prompt Learning. WACV 2024.
3.	The strategy of scaling positive examples may inadvertently include more instances of the same target images, while the scaling negative strategy treats all other images in the dataset as negatives, potentially exacerbating false-negative issues.
4.	There are some typos:
a) There is a mistake on line 299.
b) The use of " " is not formal in line 555.

**Suitability:**

3

---

### Official Review · Reviewer_YP7V · 2024-05-22

**Rating:** 6
**Confidence:** 3

**Summary:**

The article is situated within the domain of CIR (Composed Image Retrieval), which focuses on retrieving target images using a query composed of both a reference image and accompanying text.
The article addresses two challenges within the framework of contrastive learning. The lack of annotated triplets leads to fewer positive examples for the model, resulting in decreased performance. Additionally, the lack of negative examples for model training is also a problem. Their hypothesis is that best performance are obtained by methods using many counterexamples,
This papers proposes three Contributions:
    1. A data generation method to construct more positive examples based on images in the dataset.
    2. A two-step method to construct more negative examples. The first step updates the weights of both encoders (query and target encoders) to improve the model's ability to find target images, while the second step focuses on improving the representation of queries while keeping the representation of target images unchanged.
    3. The article also presents a comprehensive ablation study where performances are evaluated across various hyperparameters. The best results match those of the state-of-the-art.

**Strengths:**

1. The article is well-written, providing a clear and comprehensive overview of the current state-of-the-art to support their proposals.
    2. The authors clearly identify their contributions, which are of significant interest to the scientific community.
    3. The proposed ablation study is very interesting, highlighting the challenge of adjusting all hyperparameters based on the database context.
    4. The exhaustive experiment results effectively highlight the importance of the contributions made in the paper.
    5. Complexity of the approach is also a strength of the paper. It reduces significantly the  computation time from 20 hours to 30 minutes.

**Limitations:**

The figures and tables references should be improved, and the terms presented in Figure 4 should be redefined to assist the reader.

**Suitability:**

3

---

### Official Review · Reviewer_Assm · 2024-05-24

**Rating:** 4
**Confidence:** 4

**Summary:**

This paper examines the challenge of composed image retrieval in the context of limited positive examples in existing datasets, which is often attributed to the high cost of annotation. To address this issue, this paper presents a data generation method for constructing CIR triples using a multimodal large language model. The method functions as a plug-and-play plug-in, enabling the incorporation of CIR triples into existing methods without necessitating any alterations to their original architectures. The experiments demonstrate that the proposed plug-in can effectively enhance the retrieval accuracy of existing models. Furthermore, the method demonstrates efficacy in composed image retrieval with zero samples. Nevertheless, there are still some shortcomings in the description of the method and the analysis of the experimental results.

**Strengths:**

a.This paper proposes a method for generating CIR positive examples data based on multimodal large language models. This method generates high-quality positive examples based on image datasets only, effectively expanding the model training data.
b.This paper proposes a two-stage plug-and-play module that can be applied to existing CIR models with minimal additional overheads, without necessitating any changes to the existing models' architectures.
c.This paper investigates the effectiveness of the proposed method for CIR tasks in two scenarios: fully supervised and zero-sample. Extensive experiments are conducted on several existing CIR models.

**Limitations:**

a.As this paper proposes a plug-in that is orthogonal to the existing methods, it should be explained in detail how the method proposed in this paper is applied to the existing CIR model. Currently, there is less description of this part in the text, which should be rectified. It is recommended that the experimental section be expanded to include more detailed implementation details.
b.With regard to Table 5, the authors present the experimental results for in-domain and out-of-domain data, but do not provide a comprehensive analysis of these results. A similar issue is evident in Section 4.3, where the majority of the author's space is dedicated to merely describing the outcomes of the experiments, with comparatively less attention devoted to analysing the underlying causes of the observed experimental phenomena.

**Suitability:**

3

---

### Official Review · Reviewer_3SbY · 2024-05-24

**Rating:** 2
**Confidence:** 4

**Summary:**

This paper analyzes the origin of the retrieval performance of composed image retrieval and explores the potential of generating more positives and negatives for the existing CIR methods. To address the problem of lack of positives, the authors propose a data generation method to construct more triplets for CIR. To address the problem of lack of negatives, the authors design a plug-and-play two-stage fine-tuning method for CIR. The experiments on the general domain and fashion domain prove the effectiveness of the proposed methods.

**Strengths:**

+ The idea of constructing more positives and negatives for CIR is reasonable, and has been validated by the ablation study.
+ The analysis of negative construction is interesting, and has been little noticed in existing work.
+ The extensive experiments prove the effectiveness of the proposed method.

**Limitations:**

-	The novelty is not enough for ACM MM. Although the idea of constructing more positives and negatives is reasonable, the proposed scaling positives and scaling negatives methods are the common methods for multimodal learning, and lack novelty.
-	It seems the size of generated positives and negatives are restricted by the existing dataset. The number of generated positives equals M, while the number of negatives is restricted by the size of the candidate set.
-	At the end of sec. 3.3, the authors just discussed the detailed time of training, while does not introduce the detailed machine configuration and training set. It is meaningless to discuss time cost without a detailed experimental setup.
-	The new noun in the axis should be introduced in the caption, such as ‘SP’ and ‘SN’ in Fig. 1(b).
-	Some typos: in line 283, Fig, 2 -> Fig.~2; there is an unexpected line break in line 299. The left double quotation mark should be `` in latex.

**Suitability:**

3

---

### Meta-Review · Area_Chair_pT1z · 2024-07-02

**Recommendation:** Accept (Poster)
**Confidence:** 4

**Metareview:**

The article is dedicated to Composed Image Retrieval (CIR), it explores the potential of generating more positives and negatives examples to improve CIR within a contrastive learning framework. Globally, the 4 reviewers opinions revolve around weak/border acceptance or rejection, because they oscillate between the fact that the main idea, augmenting data to improve the framework, is not original but that the experiments prove the effectiveness of the solution, that the plug-and-play module is relevant (making the proposal applicable to other state-of-the-art techniques without structural modification), as well as the study on complexity which is a strength of the proposal. Consequently, this work has no place in an oral presentation, but it has no major drawbacks and could be presented as a poster in order to discuss with the ACM community on its contributions to current CIR architectures with low complexity.